# To Eat or Not to Eat?—Food Safety Aspects of Essential Metals in Seafood

**DOI:** 10.3390/foods12224082

**Published:** 2023-11-10

**Authors:** József Lehel, Márta Magyar, Péter Palotás, Zsolt Abonyi-Tóth, András Bartha, Péter Budai

**Affiliations:** 1Department of Food Hygiene, University of Veterinary Medicine Budapest, István u. 2., 1078 Budapest, Hungary; magyar.marti@gmail.com; 2National Laboratory for Infectious Animal Diseases, Antimicrobial Resistance, Veterinary Public Health and Food Chain Safety, University of Veterinary Medicine Budapest, István u. 2, 1078 Budapest, Hungary; 3The Fishmarket Fish Trading Company, Törökbálinti u. 23, 2040 Budaörs, Hungary; palotas.peter@thefishmarket.hu; 4Department of Biomathematics and Informatics, University of Veterinary Medicine Budapest, István u. 2., 1078 Budapest, Hungary; abonyi-toth.zsolt@univet.hu; 5Department of Animal Hygiene, Herd Health and Mobile Clinic, University of Veterinary Medicine Budapest, István u. 2., 1078 Budapest, Hungary; bartha.andras@univet.hu; 6Institute of Plant Protection, Hungarian University of Agriculture and Life Sciences, Georgikon Campus, Deák F. u. 16, 8360 Keszthely, Hungary

**Keywords:** seafood, oyster, shellfish, squid, essential metals, micronutrients, fitness for consumption

## Abstract

The popularity of seafoods is high due to their superb dietary properties and healthy composition. However, it is crucial to understand whether they adequately contribute to our essential nutritional needs. Small amounts of essential metals are indispensable in the human body to proper physiological functioning; their deficiency can manifest in various sets of symptoms that can only be eliminated with their intake during treatment or nutrition. However, the excessive consumption of metals can induce undesirable effects, or even toxicosis. Shellfish, oyster, and squid samples were collected directly from a fish market. After sample preparation, the concentration of essential metals (cobalt, chromium, copper, manganese, molybdenum, nickel, and zinc) was detected by Inductively Coupled Plasma Optical Emission Spectrometry. The results were analyzed statistically using ANOVA and two-sample *t*-tests. The average concentration of the investigated essential elements and the calculated burden based on the consumption were below the Recommended Dietary Allowances and Tolerable Upper Intake Levels. Based on these results, the trace element contents of the investigated seafoods do not cover the necessary recommended daily intake of them, but their consumption poses no health hazard due to their low levels.

## 1. Introduction

Recently, the popularity of seafoods has increased. They are becoming more and more sought after due to their superb dietary properties and healthy composition [1]. Given that seafoods take up a growing proportion of meat consumed by humans per annum, it is crucial to understand whether they adequately contribute to our essential nutritional needs [2,3].

The nutritional value of seafoods is almost the same as that of fish, primarily marine fish species with lean flesh. Generally, the flesh of fish and seafoods is mostly low in energy and fat, but (mainly crustaceans) rich in highly digestible protein and vitamins (vitamins D, E, A, and B_12_) and minerals (iron, zinc, selenium, etc.) essential for the human body [4,5,6]. For these reasons, they are not only popular in maritime countries, but are also consumed all over the world.

Due to the nutritional characteristics of aquatic animals (i.e., filter-feeding activity), live bivalve mollusks (primarily shellfish) can accumulate different contaminants and thus several metals, and may be sources of important chemical hazards relevant to food safety [7].

Of the 115 known chemical elements, only 25 are called essential. Four elements (carbon, oxygen, hydrogen, and nitrogen) play the largest role in the structure of the body, making up almost 96% of the human body [8]. The remainder can be divided into two major groups: one of the groups includes the main minerals (macrominerals, macronutrients), while the other includes trace elements (microminerals, micronutrients). Macronutrients are those that are present in amounts greater than 0.005% of the body weight and are essential at levels of 50 mg or more per day, and micronutrients are those that are present in amounts less than 0.005% of the body weight and are essential at a concentration of <50 mg per day [9]. Calcium (Ca), magnesium (Mg), potassium (K), sodium (Na), chloride (Cl), phosphorus (P) and sulfur (S) belong to the main minerals. Iodine (I), zinc (Zn), selenium (Se), iron (Fe), manganese (Mn), copper (Cu), cobalt (Co), molybdenum (Mo), fluorine (F), chromium (Cr) and boron (B) are trace elements [10,11]. Trace elements can be divided into essential and non-essential elements based on their biological importance. Essential micronutrients have been shown to be indispensable for the balanced functioning of the human body, while the biological role of non-essentials is still unclear [12,13,14,15,16]. Trace elements include a wide variety of metals, and the positive and negative effects of cobalt, chromium, copper, manganese, molybdenum, nickel, and zinc are discussed in this study.

Small amounts of essential metals are indispensable to the proper physiological functioning of the human body. Their deficiency can manifest in various sets of symptoms that can only be eliminated with the intake of those metals that are lacking during treatment or nutrition. However, the excessive consumption of metals can induce undesirable effects or even toxicosis [17,18,19].

Cobalt is an essential trace metal for the human body; it is the central atom of cobalamin (vitamin B_12_) and it has a role in the regulation of blood pressure [19], while laos being active in the hemopoietic system, including the production of red blood cells, platelets, DNA, fatty acid synthesis, and energy production [20]. Cobalt is a cofactor of different enzymes involved in the biosynthesis of DNA and in the metabolism of amino acids in humans [21], and it is necessary to the normal functioning of the thyroid gland [22]. It can thermodynamically reduce the harmful effects of reactive oxygen species (ROS) [23].

An insufficient supply of cobalt manifests as a lack of vitamin B_12_, so its symptoms are very diverse. Hematological, gastrointestinal, psychiatric, and neurological changes may occur in humans [24].

In the case of excessive intake, goiter, hypothyroidism, polycythemia [25,26], congestive heart failure and anemia may occur in humans [27]. The main adverse effects associated with excessive exposure to Co are neurological, cardiovascular, and endocrine effects in humans [28].

Chromium participates in carbohydrate, protein, and lipid metabolism; however, its exact role and mechanism of action in the human body are not yet fully understood [29,30]. It may play a role in maintaining the configuration of RNA molecules, since chromium has a proven role in collagen cross-linking [31]. Furthermore, chromium is a component of several enzyme systems; out of these, one is important, that is, GTF (glucose tolerance factor), which is an active component [32]. GTF can bind to insulin and thus activates it and potentiates its effect [25]. It promotes the growth and reduces blood pressure and cholesterol level. Its deficiency can lead to atherosclerosis [30].

Chromium deficiency can cause growth disorders in children, especially those suffering from protein calorie malnutrition. In its absence, insulin resistance and glucose intolerance may develop, and the insulin sensitivity of peripheral tissues and the rate of glucose removal decrease, which is also supported by animal experiments. Insufficient chromium intake in rats reduced the lifespan of the animals and caused corneal lesions [33,34,35].

Chromium occurs in two stable oxidation states, Cr^3+^ and Cr^6+^, which differ significantly in toxicity. Commonly, trivalent chromium compounds are much less toxic than hexavalent chromium compounds. Cr^6+^ compounds can induce different adverse effects such as irritation of the skin and eye, disturb the function of respiratory, gastrointestinal, reproductive, and immune systems, and cause hematological disorders and tumors in humans [36].

Copper is a component of important enzymes (cytochrome c oxidase, amino oxidase, catalase, peroxidase, ascorbic acid oxidase, cytochrome oxidase, plasma monoamine oxidase, erythrocuprein (ceruloplasmin), lactase, uricase, tyrosinase, superoxide dismutase, etc.) [37]. It is essential to the function of the hematological and nervous systems, the formation and growth of bones and the creation of the myelin sheath [38]. It promotes the absorption of iron from the gastrointestinal tract, and the incorporation of it into the hemoglobin [25,39].

Symptoms of copper deficiency include anemia (hypochromic, microcytic), skeletal abnormalities, neonatal ataxia, depigmentation, abnormal hair and fur growth, heart failure, and digestive disorders. There are also abnormalities in growth and reproduction. Copper deficiency can cause cardiac hypertrophy and sudden cardiac arrest [25,40].

Copper poisoning is rare and usually occurs secondary to Wilson’s disease in humans. In this disease, large amounts of copper are deposited in the liver, brain, and other sites [41]. Excessive dietary intake also causes deposits in the liver, which reduce the concentration of hemoglobin and hematocrit in the blood. As a result of acute copper toxicosis, the destruction of the epithelial layer of the GI tract, hepatocellular necrosis and kidney tubular necrosis can be manifested in humans and animals [42].

Manganese is an essential trace element for cellular metabolic processes. It is a cofactor of important enzymes (such as Mn superoxide dismutase, glutamine synthetase, arginase and pyruvate carboxylase) that play a role in cellular metabolism, antioxidant defense, neurological function, normal growth of bones and connective tissue, immune function, and reproduction [43,44].

The symptoms of rare human manganese deficiency are skin inflammation, reduced nail and hair growth, elevated serum calcium and phosphorus concentrations, increased alkaline phosphatase activity, decreased serum cholesterol levels and reductions in clotting protein levels [45].

During excessive oral intake, Mn is absorbed from the GI tract and transported to the liver, kidneys, pancreas, bones, and brain, where it can be concentrated [46]. Manganese can cross the blood–brain barrier (BBB) via various transport processes, such as active transport or facilitated diffusion, as a result of which the brain is the major target organ of manganese poisoning [45,47]. Mn is mainly excreted in the intestine via bile and to a lesser extent through the kidneys in urine. The major symptoms of human Mn poisoning include hypermanganesemia, liver cirrhosis, impair cardiovascular function, polycythemia, dystonia, and parkinsonism-like symptoms [44].

As a result of the toxic effect of manganese at the molecular level, the production of reactive oxygen species increases, the functioning of mitochondria changes, stress occurs in the endoplasmic reticulum, apoptotic processes are initiated and homeostasis of other trace elements (cobalt, nickel, zinc) is disrupted [43,44,47].

Skeletal deformities are described in manganese-deficient animals. Manganese deficiency can induce longitudinal growth in the tibia, which may be due to the role of manganese in cell division, differentiation, and apoptosis. Furthermore, it can lead to the inhibition of chondrocytes proliferation and increase chondrocyte apoptosis in chicks [48].

Molybdenum is a component of many metalloenzymes, including xanthine oxidoreductase (xanthine oxidase), aldehyde oxidase, nitrate reductase and hydrogenase. Through the enzymes xanthine oxidase and aldehyde oxidase, it plays a role in iron utilization and electron transport. Xanthine oxidase is actively involved in the uptake and release of iron in the form of ferritin through the intestinal mucosa, as well as in the liver, placenta and hematopoietic organs [25,49]. Molybdenum is a cofactor for enzymes involved in the formation of uric acid, in the metabolism of sulfur-containing amino acids and nitrogen-containing compounds found in DNA and RNA, and in the oxidation and detoxification of many other compounds [50]. Due to its deficiency, high blood sulfite and urate levels are detected [51], which can lead to gout. Its low intake is also a predisposing factor in the formation of kidney stones [50].

Acute poisoning has not yet been described in humans, but animal experiments have shown that a chronically high intake of molybdenum (above 10 mg/day) causes diarrhea, growth disorders, infertility and low birth weight, and has harmful effects on the lungs, kidneys, and liver [52,53].

Nickel is distributed thoroughly in the body, reaching higher levels in the kidneys, bones, and lungs. It is essential to the synthesis of different enzymes, e.g., hydrogenase and carbon monoxide dehydrogenase, and it is found in several enzymes, e.g., nickel-tetrapyrrole coenzyme, Cofactor F430, methyl coenzyme M reductase, etc. Thus, it plays an important role in living systems [54,55,56].

Nickel deficiency is relatively rare in humans because nickel is found throughout the environment, and thus in different types of food. However, its deficiency can induce growth retardation in embryo, severe allergic reaction, hepatitis, pulmonitis, hematological disorders (e.g., anemia), paralysis, skin rashes, and infection of the urinary tract in humans [54,57,58].

Animal experiments showed that nickel deficiency can cause increased mortality, decreased body weight gain and changes in blood parameters in lambs; it can also result in leg abnormalities, skin dermatitis and changes of liver lipids in chicks, and induce reproductive problems in swine (e.g., delayed estrus, increased neonatal mortality) [59,60].

A high amount of nickel can induce allergic reactions, gastrointestinal disorders (nausea, vomiting, abdominal discomfort, and diarrhea), hematological problems (high red blood cell counts), nephrotoxic disorders (edema, hyperemia, parenchyma degeneration), reproductive problems (embryotoxic and teratogenic effect), and lung cancer in humans, and can disturb the functions of the respiratory tract, cardiovascular, nervous, and immune systems [57,58,61].

The excessive intake of nickel is toxic and carcinogenic, and causes weight loss, heart and liver damage, and skin irritation in mice [62].

Zinc is a cofactor or component of many enzymes, including lactate dehydrogenase, alcohol dehydrogenase, glutamate dehydrogenase, alkaline phosphatase, carbonic anhydrase, carboxypeptidase, superoxide dismutase, retinin reductase, and DNA and RNA polymerase. Through them, zinc takes part in the metabolism of nutrients and in the replication of cells [21,63]. It also plays a role in gene expression and the metabolism of nucleic acids and amino acids. The bioavailability and metabolism of vitamins A and E largely depends on the body’s zinc supply [64]. Zinc is needed for the proper development of testicles in animals [42], and for the functioning of taste buds. It is essential for tissue repair processes and wound healing, plays a vital role in protein synthesis and protein digestion, and is necessary for the proper functioning of insulin, as it is an essential component. However, it is also an important part of plasma [25,39]. It is essential for growth, hormonal interactions, and the proper functioning of the immune system [65].

Its deficiency can cause hypogonadism, growth depression, impaired wound healing, a decreased ability to taste and smell, parakeratosis, and Brandt’s syndrome (*Acrodermatitis enteropathica*) [25].

High zinc levels cause pancreatitis, anemia, muscle pain, and acute renal failure in humans [66,67]. In addition, they can cause gastrointestinal irritation, vomiting, reduced immune functions, and a decrease in HDL (high density lipoprotein) cholesterol levels in rats [68].

In the course of our work, the concentrations of some essential metals accumulated in the edible tissues of various seafoods were determined to obtain answers to the following questions: (1) Is there a difference in the concentrations of essential metals in the edible tissues of each aquatic mollusk species (mussels, oysters, squid)?; (2) To what extent do trace elements derived from seafood cover our daily needs?; (3) Can the consumption of squid, oysters and mussels exceed the maximum daily intake without compromising health?

## 2. Materials and Methods

### 2.1. Sampling

Samples of mollusks were collected from a local fishery product market in Hungary weekly for 20 weeks. The seafood originated from Denmark, Italy (shellfish), France (oysters), and Argentina (squids). During the study, 42 shellfish, including black mussel (*Mytilus galloprovincialis*), blue mussel (*M. edulis*), Vongole (*Venerupis philippinarum*), and Amanda cockle (*Glycymeris glycymeris*), 34 oysters (Pacific oyster (*Crassostrea gigas*), Portuguese oysters (*C. angulate*)), and 38 squids (European squid (*Loligo vulgaris*)) were investigated to determine their essential metals contents, such as cobalt, chromium, copper, manganese, molybdenum, nickel, and zinc.

### 2.2. Preparation of Samples

The soft tissues (whole body and all tissues) of each shellfish species and oyster species were taken out with a plastic tool after opening the shells. After cutting and homogenization (Potter S, B. Braun Biotech International GmbH, Melsungen, Germany), the samples were placed into a properly labeled plastic bag. The tubes of squids (without the gastrointestinal tract) were prepared similarly. After preparation the samples were stored in a freezer (So-Low Ultra-Low Freezer, Model C85-9, Environmental Equipment Co. Inc., Cincinnati, OH, USA) at −70 °C until the analysis. At the start of the analytical procedure 0.5 g from each sample was weighed into a CEM MARS XPreSS teflon vessel (CEM, Matthews, NC, USA) for sample digestion in the case of all species of mollusks. Then they were decomposed by 5 mL of nitric acid and hydrogen peroxide in a CEM MARS6 microwave digestion system (CEM Corporation, Matthews, NC, USA) (Ramp: 35 min; temperature: 200 °C; hold: 50 min; energy: 1700 W). Then, the sample was made up to 25 mL with deionized water and analyzed by an Inductively Coupled Plasma Optical Emission Spectrometer (ICP-OES) after double dilution with deionized water. The blank and the QC samples were prepared in the same way.

### 2.3. Chemicals and Standards

Nitric acid (69 m/m%, Aristar; VWR International Ltd., Leicestershire, UK) and hydrogen peroxide (30% m/m%, Normapur; VWR International Ltd., Leicestershire, UK) were used with trace analysis for sample preparation. All laboratory glassware and plastic tools were cleaned with 0.15 M hydrochloric acid solution (37 m/m%, Aristar; VWR International Ltd., Leicestershire, UK) and then rinsed with deionized water produced by a Purite Select Fusion 160 BP water purification system (Suez Water Ltd., Thame, UK). Calibration was performed by the application of ICP multi- (Perkin Elmer Inc., Shelton, CT, USA) and mono-element (VWR International Ltd., Leicestershire, UK) standards for quantitative ICP measurement. Argon gas of 4.6 purity was used for measurements (Messer Hungarogáz Ltd., Budapest, Hungary). Quality control (QC) standards were prepared from a certified reference material of mussel tissue (ERM CE278k, European Commission, Joint Research Centre, Geel, Belgium).

### 2.4. Analytical Method

The concentrations of essential metals were determined using an Inductively Coupled Plasma Optical Emission Spectrometer (ICP-OES) instrument (Perkin Elmer Optima 8300 DV, Perkin Elmer, Shelton, CT, USA). During the analytical procedure, the following measurement parameters were applied: RF generator—40 MHz solid state, free running, flat plate plasma technology; RF power—1300 W; nebulizer type—BURGENER PEEK MIRA MIST; Plasma gas flow rate—12 dm^3^/min; auxiliary gas flow rate—0.2 dm^3^/min; nebulizer gas flow rate—0.7 dm^3^/min; observation height—15 mm. The calibration curves were between 0 and 200 mg/kg. The limit of detection (LOD) was 0.05 mg/kg for cobalt, chromium, copper, manganese, and zinc, 0.5 mg/kg for molybdenum, and 0.2 mg/kg for nickel.

Internal quality control of the measurements was carried out by measuring QC samples of known metal concentration at least 10 times (ERM CE278k, mussel tissue). After discarding the extremes, the standard deviations of data (SD) were established, all of which had to remain within ±15% of the nominal concentration value in order for their QC measurement to be accepted (Table 1). All sample, calibration and blank solutions were analyzed in 3 replicates.

### 2.5. Validation of the Method

Several validation parameters were established according to the relevant guidelines to assess the reliability of the analytical method and sample preparation [69]. Limits of detection (LOD) and limits of quantitation (LOQ) were defined as three and ten times the standard deviation of the signals of the blank samples, respectively. Trueness was determined by analyzing the ERM-CE278k-certified reference material (CRM). Spiked samples of the CRM were used for those elements that did not have certified values. The precision was determined as the relative standard deviation of the concentration values of ten replicates of the same CRM or spiked CRM samples. Percentages were used to express both precision and trueness. The matrix effect was not studied separately as the internal standard solution used in each sample compensated for this effect. Precision was accepted if the deviation of the measured parameter did not exceed 15%, and trueness values were accepted below ±20%. The recoveries of the quality control standards were between 93.4 and 104.7%.

### 2.6. Statistical Analysis

The statistical analysis of the measured concentrations was performed using the R statistical software (version 3.1.3.) and the statistical program of Microsoft Excel. The quantity of each essential element was compared using an ANOVA test among the shellfish (mussel) and oyster species, and amongst the squids. Where the conditions of the ANOVA test were not met, the logarithm of the measured quantities was calculated, so the test could be performed. In order to prepare the statistics correctly, the concentrations of the metals below the limit of detection were replaced by the LOD value [70].

During the assessment of the possible intake of essential elements, the concentrations of the metals detected in the samples and an average daily consumption of 200 g were the basis. Then, this daily burden was divided by an average consumer’s body weight (60 kg).

The average concentrations of the investigated essential elements and the calculated burden based on the consumption were compared to the Recommended Dietary Allowances and Tolerable Upper Intake Levels stated by EFSA and the Institute of Medicine of the United States of America.

## 3. Results

The average metal contents of the seafoods studied (mussels, oysters, and squid) are shown in Table 2.

During the analysis of the samples, the concentrations of manganese, copper and zinc were above the detection limit in the squid, oyster, and mussel samples. However, for cobalt, chromium, molybdenum and nickel, the proportion of values below the limit of measurement was above 50%.

The cobalt content of mussels was significantly higher than those of squid and oysters (*p* < 0.001).

The chromium concentration of the mussel samples was significantly higher than that of the oysters (*p* = 0.004).

The copper content of oysters (16.46 mg/kg) was significantly higher (*p* < 0.001) than that of mussels (1.16 mg/kg) and squid (7.20 mg/kg).

The concentration of manganese in oysters (4.88 mg/kg) was significantly higher (*p* < 0.01) compared with squid (0.29 mg/kg) and mussel (1.65 mg/kg).

There was no significant difference (*p* > 0.1) between the molybdenum contents of the squid, oyster and mussel samples examined.

The nickel concentration in the shellfish samples was significantly higher than that of the squid (*p* < 0.001).

Examining the zinc content, the three groups differed significantly; it was lowest in squid (11.31 mg/kg), while that of oysters (202.60 mg/kg) was the highest (*p* < 0.001).

The upper and lower limits of the absorption and consumption values (intervals), calculated from the metal contents of the shell, oyster and squid samples, are displayed in Table 3. For the Recommended Daily Allowance (RDA), the recommendations set by the European Food Safety Authority (EFSA) and the National Institutes of Health (NIH) are included. The Tolerable Upper Intake Level (UL) is based on the values described by the NIH [71].

## 4. Discussion

According to a study on consumer habits in the European Union, the average consumer consumes 15–30 kg of seafood a year, which is high [72]. Hungarian consumption is lower than this, but shows an increasing trend. While the average consumption was still 5 kg in 2009, it is projected to increase. A positive correlation can be found between seafood consumption and age. The over-44 age group has the highest number of consumers, while the younger demographic group (15–24 years) has the highest proportion of those who do not consume such products.

### 4.1. Cobalt

The average concentration of cobalt in squid was 0.05, 0.05 and 0.16 mg/kg wet weight (w.w.), and the calculated metal uptake was 0.17, 0.17 and 0.53 µg/kg in squid, oysters and mussels, respectively.

The measured Cobalt levels were similar to those detected by Sivaperumal et al., with a cobalt concentration of 0.05–0.85 mg/kg [73]. However, an even higher cobalt concentration was noted in the *Donax* mussel species (*Donax venustus*, *D. trunculus*) (1.02–2.86 mg/kg) living off the west coast of Egypt, and in *Gafrarium pectinatum* (4.25 mg/kg), which is five times that seen in the measurement of Sivaperumal et al. [74].

Similarly, the average amount of cobalt was 0.09 ± 0.11 µg/g in mollusk spp. (octopus (*Octopus vulgari*s), clam (*Mytilus galloprovincialis*), shrimp (*Caridina* sp.), squid (*Loligo farbesi*), mussel (*Mytella guianensis*)) collected from different regions of Brazil [75].

The recommended daily intake of cobalt is 0.17 µg [76]. In our study, the average metal intake calculated from the results covers 100% of the recommended daily intake for squid and oysters and 29.4% for mussels.

### 4.2. Chromium

The detected average concentration of chromium was 0.06, 0.05 and 0.11 mg/kg w.w., and the calculated metal burden was 0.20, 0.17 and 0.38 µg/kg in squids, oysters and mussels, respectively.

On the west coast of Egypt, the chromium content was very high, 3.72–18.58 mg/kg, with the highest value measured in shellfish species (*Donax venustus*) [74]. In measurements in the Gulf of Suez, the chromium concentration ranged from 2.34 to 7.99 mg/kg in mollusk (*Barbatus barbatus*); this value is more six times greater than our measurements [77].

Lower, but still almost 3 times higher, values were measured in our results in mussel species (*Chamelea gallina*) living in the waters of southern Spain, where chromium was present in a concentration of 0.24–1.22 mg/kg [78]. In a study by De Mora et al., the amount of chromium in mollusks in the Persian Gulf ranged from 0.01 to 3.76 mg/kg [79]. Similar values were measured by Sivaperumal et al. in a study of mollusks purchased in India, with chromium concentrations ranging from 0.18 to 3.65 mg/g [73].

Similar results were described for different European regions. The mean concentration of chromium was 0.31 ± 0.03 mg/kg w.w. in mussel spp. (*Donax trunculus*) around Sicily, Italy, originating from the Ionian Sea [80].

The detected concentration of chromium was 0.15 ± 0.11 mg/kg w.w. in prawn spp. (*Pandalus borealis*) in Greenland [81].

The average concentration of chromium was 0.40 ± 0.03 mg/kg in different cephalopods, including octopus spp. (spider octopus (*Octopus salutii*), common octopus (*O. vulgaris*)), cuttlefish spp. (common cuttlefish (*Sepia officinalis*), pink cuttlefish (*S. orbihnyana*), elegant cuttlefish (*S. elegans*)) and squid spp. (European squid (*Loligo vulgaris*), broadtail squid (*Illex coindetii*)) collected from the Mediterranean Sea [82].

The recommended daily chromium intake is 20 µg [83]. For the squid samples, the calculated metal uptake covers only 4.9% of our chromium requirement, and the chromium content of oysters is 0.85%. For mussels, the metal content covers 1.9% of daily demand.

### 4.3. Copper

The measured average amount of copper was 7.20, 16.46 and 1.16 mg/kg w.w., and the calculated metal uptake was 23.97, 54.85 and 3. 86 µg/kg in squid, oyster and mussel species, respectively.

Similar results were obtained in the research of Hamed and Emara [77]. The copper content of mollusk species (*Barbatus barbatus*) in the Gulf of Suez (Red Sea) was 1.6–12.17 mg/kg. El Nemr et al. measured a copper content of 1.54–11.8 mg/kg in different shellfish species living on the west coast of Egypt (*Tapes decussata*, *Paphia undulata*, *Venerupis decussata*, *Gafrarium pectinatum*, *Donax* species) [74].

Concentrations much higher than our measurement results were measured in the Gulf of Oman, where the samples contained 60.9–210 mg/kg copper [79]. In contrast, Olmedo et al. measured a copper concentration of 0.918–2.314 mg/kg lower than the values we measured in frozen mussels from Andalusia (southern Spain) [84].

Similarly, lower results were detected in different European regions and South America. The concentrations of copper were 8.00 ± 0.16, 0.68 ± 0.001, and 8.78 ± 0.47 µg/g w.w. in octopus (*Octopus vulgaris*), mussel (*Mytilus edulis*), and shrimp (*Litopenaeus vannamei*) collected from Spanish, Dutch, and Portuguese fishery markets [85].

The detected concentration of copper was 5.0 ± 1.0 mg/kg w.w. in prawn spp. (*Pandalus borealis*) in Greenland [81].

The mean copper level was 17.39 ± 11.73 µg/g in mollusk spp. (octopus (*Octopus vulgaris*), clam (*Mytilus galloprovincialis*), shrimp (*Caridina* sp.), squid (*Loligo farbesi*), mussel (*Mytella guianensis*)) collected from different regions of Brazil [75].

The mean level of copper was 23.77 ± 10.29 mg/kg in different cephalopods, including octopus spp. (spider octopus (*Octopus salutii*), common octopus (*O. vulgaris*)), cuttlefish spp. (common cuttlefish (*Sepia officinalis*), pink cuttlefish (*S. orbihnyana*), elegant cuttlefish (*S. elegans*)), and squid spp. (European squid (*Loligo vulgaris*), broadtail squid (*Illex coindetii*)) collected from the Mediterranean Sea [82].

Amiard et al. (2008) measured higher concentrations of copper, such as 58, 78 and 141 mg/kg w.w., in oyster spp. (*Crassostrea gigas*, *Ostrea edulis*, *Saccostrea cucullate*) originating from the coastal areas of China, France, and England due to the anthropogenic contamination of the environment. However, the detected levels were 1.3 and 1.1 mg/kg w.w. in green mussels (*Perna viridis*) and clams (*Marcia hiantina*) in the samples collected from cleaner and safer areas of China [86].

The recommended daily intake for copper is 1600 µg for men and 1300 µg for women [87]. Based on the calculated metal uptake, the amount of metal introduced covers a small part of the daily copper need: in the case of squid, 1.5% of the needs of men and 1.8% of the needs of women; for oysters, 3.4% of men’s needs and 4.2% of women’s needs; concerning mollusks, 0.24% of men’s and 0.3% of women’s needs.

### 4.4. Manganese

The measured average manganese concentrations were 0.29, 4.88 and 1.65 mg/kg w.w., and the calculated metal uptake was 0.96, 16.25 and 5.49 µg/kg in squids, oysters and mussels, respectively.

Olmedo et al. examined manganese levels of 0.032–0.104 mg/kg, similar to our values, in squid samples (*Dosidicus gigas*) from Argentina [88]. Frozen mussel samples from southern Spain and Italy (*Mytilus edulis*, *Venus gallina*) had a manganese content of 0.114–0.928 mg/kg lower than our values.

Measurements by the Food Standards Agency showed a lower manganese concentration of 0.730 mg/kg in frozen mollusk samples than in our oyster and mussel samples [88]. Similar data were obtained in the study of canned mussels purchased in the Canary Islands, where manganese levels of 1.8 mg/kg were detected, and in the aquatic life of the Magellan Strait, Chile, where a manganese concentration of 1.46 mg/kg was measured [89,90].

Similar findings were found at different European regions and in South America. The concentrations of manganese were 0.68 ± 0.02, 2.82 ± 0.04, and 0.93 ± 0.07 µg/g w.w. in octopus (*Octopus vulgaris*), mussel (*Mytilus edulis*), and shrimp (*Litopenaeus vannamei*) collected from Spanish, Dutch, and Portuguese fishery markets [85].

The average level of manganese was 3.23 ± 0.49 mg/kg w.w. in mussel spp. (*Donax trunculus*) from around Sicily, Italy, originating from the Ionian Sea [80].

The average concentration of manganese was 1.39 ± 0.99 µg/g in mollusk spp. (octopus (*Octopus vulgaris*), clam (*Mytilus galloprovincialis*), shrimp (*Caridina* sp.), squid (*Loligo farbesi*), mussel (*Mytella guianensis*)) collected from different regions of Brazil [75].

The National Institutes of Health recommends a daily intake of 2300 µg for men and 1800 µg for women [71]. The metal concentration measured in our samples covers only a small part of this, concerning squid—0.013% for men, 0.016% for women; oysters—0.7% for men and 0.9% for women; mussels—0.24% for men and 0.31% for women.

### 4.5. Molybdenum

The average molybdenum concentration was 0.05, 0.52 and 0.55 mg/kg w.w. detected in squids, oysters and mussels, respectively, and the metal burden induced by them was 0.17, 1.72 and 1.83 µg/kg.

In their research in the Bay of Biscay in Western Europe, Chaillou et al. measured higher molybdenum concentrations in mussels than were found in our results, ranging from 0.65 to 3.36 mg/kg [91]. Ravera et al. measured different mollusk species (*Unio mancus*, *Anodonta cygnea*, *Dreissena polymorpha*) originating from the coast of Northern Italy and found an average molybdenum content of 1.7 mg/kg, which is 3 times higher than in our results [92].

The mean level of molybdenum was 0.19 ± 0.20 µg/g in mollusk spp. (octopus (*Octopus vulgaris*), clam (*Mytilus galloprovincialis*), shrimp (*Caridina* sp.), squid (*Loligo farbesi*), mussel (*Mytella guianensis*)) collected from different regions of Brazil [75].

The recommended daily intake for molybdenum is 65 µg per day [71]. Based on the samples examined, consumption of 200 g covers a maximum of 0.28% for squid, 2.6% for oysters and 2.8% for mussels.

### 4.6. Nickel

The average amount of nickel was 0.21, 0.23 and 0.51 mg/kg w.w., and the calculated metal burden was 0.70, 0.75 and 1.69 µg/kg in squid, oyster and mussel species, respectively.

A similar result to what was found in our measurements was obtained by Sivaperumal et al., who studied mussels living in Indian waters [73]. The nickel concentration was low, with a maximum measured value of 0.89 mg/kg. Slightly higher nickel contents were measured in the Queshm Islands, where metal concentrations ranged from 1.66 to 3.02 mg/kg when tested on oysters (*Saccostrea cuculata*) [93].

Significantly higher values were measured on the west coast of Egypt, where in the study of different mollusc species (*Tapes decussata*, *Paphia undulata*, *Venerupis decussata*, *Gafrarium pectinatum*, *Donax* species) the nickel concentration ranged from 3.42 to 15.02 mg/kg [74]. However, Usero et al. detected an even higher metal content of 66.0–92.0 mg/kg when studying mussels (*Chamelea gallina*) living in the waters of southern Spain [78].

Similarly to our results, the average concentration of nickel was 0.20 ± 0.03 mg/kg w.w. in mussel spp. (*Donax trunculus*) from around Sicily, Italy, originating from the Ionian Sea [80].

A higher nickel concentration (2.12 ± 0.97 mg/kg) was detected in different cephalopods, including octopus spp. (spider octopus (*Octopus salutii*), common octopus (*O. vulgaris*)), cuttlefish spp. (common cuttlefish (*Sepia officinalis*), pink cuttlefish (*S. orbihnyana*), elegant cuttlefish (*S. elegans*)), and squid spp. (European squid (*Loligo vulgaris*), broadtail squid (*Illex coindetii*)) collected from the Mediterranean Sea [82].

The recommended daily intake for nickel is 100 µg [94]. Based on the calculation of metal uptake using the average nickel concentration of our samples, the calculated quantities cover 0.7% of the daily demand via squid consumption, 0.75% of the demand via oyster consumption and 1.7% via mussels.

### 4.7. Zinc

The average zinc concentration measured in squid, oyster and mussel species was 11.32, 202.60 and 24.96 mg/kg w.w., and the calculated metal intake related to consumption of them was 37.73, 675.33 and 83.17 µg/kg, respectively.

Similarly to our results, El Nemr et al. measured the concentration of zinc to be between 8.35 and 66.05 mg/kg in different mussel species (*Tapes decussata*, *Paphia undulata*, *Venerupis decussata*, *Gafrarium pectinatum*, *Donax* species) living on the west coast of Egypt [74]. In measurements taken at Favignana Island in Sicily, the zinc ranged from 5 to 31 mg/kg in mollusks [95]. In the research of Widdows et al., the metal concentration of the mussel species (*Mytilus galloprovincialis*) ranged from 82 to 185 mg/kg [96]. A similar zinc content was measured by Hamed and Emara in their research in the Red Sea, where the zinc concentration measured in mollusks (*Barbatus barbatus*) was 56.5–191.4 mg/kg [77].

Conti and Cecchetti, who measured values higher than the zinc content of our mussel samples, found in their study on the different mussels of the Gulf of Venice (*Mytilus galloprovincialis*, *Monodonta turbinata*) a metal content ranging from 98 to 152 mg/kg [97].

Similar results were detected in mussel, squid, and shrimp spp. in different European regions and South America. The concentrations of zinc were 14.5 ± 0.3, 11.1 ± 0.2, and 18.3 ± 0.4 µg/g w.w. in octopus (*Octopus vulgaris*), mussel (*Mytilus edulis*), and shrimp (*Litopenaeus vannamei*) collected from Spanish, Dutch, and Portuguese fishery markets [85].

The mean level of zinc was 9.11 ± 0.98 mg/kg w.w. in mussel spp. (*Donax trunculus*) from around Sicily, Italy, originating from the Ionian Sea [80].

The detected concentration of zinc was 11.0 ± 2.00 mg/kg w.w. in prawn spp. (*Pandalus borealis*) in Greenland [81].

The average detected quantity of zinc was 46.0 ± 21.0 µg/g in mollusc spp. (octopus (*Octopus vulgaris*), clam (*Mytilus galloprovincialis*), shrimp (*Caridina* sp.), squid (*Loligo farbesi*), mussel (*Mytella guianensis*)) collected at different regions of Brazil [75].

The detected mean concentration of zinc was 33.03 ± 7.84 mg/kg in different cephalopods, including octopus spp. (spider octopus (*Octopus salutii*), common octopus (*O. vulgaris*)), cuttlefish spp. (common cuttlefish (*Sepia officinalis*), pink cuttlefish (*S. orbihnyana*), elegant cuttlefish (*S. elegans*)), and squid spp. (European squid (*Loligo vulgaris*), broadtail squid (*Illex coindetii*)) collected from the Mediterranean Sea [82].

However, Amiard et al. (2008) measured higher concentrations of zinc, such as 217, 500 and 618 mg/kg w.w. in oyster spp. (*Crassostrea gigas*, *Ostrea edulis*, *Saccostrea cucullata*) originating from the coastal areas of China, France, and England due to the anthropogenic contamination of the environment. On the contrary, the levels detected were 4.8 and 5.4 mg/kg w.w. in green mussels (*Perna viridis*) and clams (*Marcia hiantina*) in the samples collected from the cleaner and safer areas of China [86].

The National Institutes of Health recommends an adequate daily zinc intake of 10.1 mg for men and 8.2 mg for women [71]. Of this, the metal consumption calculated from the metal content of our squid samples covers 0.37% of the demand for men and 0.46% of the demand for women. Oysters provide 6.7% of the demand for men and 8.24% for women. With shellfish consumption, men can cover 0.82% of their daily needs and women 1.01%.

Based on the presented results, it can be stated that the average concentrations of the examined essential elements and the calculated burden based on the consumption only covered the daily necessity in the case of cobalt when consuming the investigated mollusk species (squid, oyster, mussel). Concerning chromium, copper, manganese, molybdenum, nickel, and zinc, the calculated intakes were below the Recommended Dietary Allowances and Tolerable Upper Intake Levels.

The trace element contents of investigated seafood samples do not cover the recommended daily intake of essential metals, but due to their low concentrations, the examined seafoods pose no particular health hazard.

## Figures and Tables

**Table 1 foods-12-04082-t001:** Results of quality control (QC) measurement using certified reference material (mussel tissue, ERM-CE287k).

Element	Certified Value (mg/kg)	Measured Value (mg/kg)	Measured with Spike * (mg/kg)	LOD (mg/kg)	Recovery (%)
Chromium	0.73	0.67 ± 0.01	NA	0.05	91.0
Cobalt	0.21	0.22 ± 0.02	NA	0.05	104.7
Copper	5.98	5.70 ± 0.14	NA	0.05	115.2
Manganese	4.88	4.56 ± 0.05	NA	0.05	93.4
Molybdenum	ND	0.43 ± 0.01	2.86 ± 0.04	0.5	97.0
Nickel	0.69	0.67 ± 0.02	NA	0.2	97.1
Zinc	71.00	70.90 ± 0.59	NA	0.05	99.8

* Spiked samples were used for those elements that did not have certified values in ERM-CE278k. The spiked values were equal to 2.5 mg/kg; NA = not applicable; ND = no data.

**Table 2 foods-12-04082-t002:** Trace element content of the tested species (mean, mg/kg wet weight).

Metal	LOD (mg/kg)	Squid			Oyster			Mussel		
		n	Mean	SE	n	Mean	SE	n	Mean	SE
Cobalt	0.05	19	0.05 a	0.00	17	0.05 a	0.00	21	0.16 b	0.11
Chromium	0.05	19	0.06 ab	0.02	17	0.05 a	0.00	21	0.11 b	0.14
Copper	0.05	19	7.20 b	4.24	17	16.46 c	9.87	21	1.16 a	0.53
Manganese	0.05	19	0.29 a	0.12	17	4.88 c	2.26	21	1.65 b	1.23
Molybdenum	0.50	38	0.05 a	0.00	34	0.52 a	0.07	42	0.55 a	0.20
Nickel	0.20	38	0.21 a	0.04	34	0.23 a	0.05	42	0.51 b	0.65
Zinc	0.05	38	11.32 a	1.60	34	202.60 c	88.41	42	24.96 b	20.04

n = number of samples; LOD = limit of detection; SE = standard Error. Different lowercase letters (a–c) in each column indicate significant differences between species for each element (One-way ANOVA, *p* < 0.05).

**Table 3 foods-12-04082-t003:** Average metal uptake calculated from metal concentrations measured in samples.

Metal		Measured Quantity (mg/kg)	SE	Average Metal Uptake (µg/kg) (Interval)	SE	RDA (µg/day)	UL (mg/day)
Cobalt	Squid	0.05		0.17	0.00	0.17	ND
	Oyster	0.05	0.00	0.17 (0.16–0.17)	0.00		
	Mussel	0.16	0.11	0.53 (0.17–1.98)	0.37		
Chromium	Squid	0.06	0.02	0.20 (0.02–0.50)	0.08	20	ND
	Oyster	0.05	0.00	0.17 (0.17–0.25)	0.01		
	Mussel	0.11	0143	0.38 (0.20–2.19)	0.49		
Copper	Squid	7.20	4.24	23.97 (4.20–81.44)	14.15	Male: 1600 Female: 1300	5
	Oyster	16.46	9.87	54.85 (17.43–147.35)	32.90		
	Mussel	1.16	0.53	3.86 (0.74–8.54)	1.78		
Manganese	Squid	0.29	0.12	0.96 (0.41–2.12)	0.40	Male: 2300 Female: 1800	11
	Oyster	4.88	2.26	16.25 (6.34–33.94)	7.52		
	Mussel	1.65	1.23	5.49 (1.16–20.18)	4.10		
Molybdenum	Squid	0.05	0.00	0.17	0.00	65	2
	Oyster	0.52	0.07	1.72 (1.67–2.89)	0.22		
	Mussel	0.55	0.20	1.83 (2.33–4.85)	0.67		
Nickel	Squid	0.21	0.04	0.70 (0.67–1.26)	0.12	100	1
	Oyster	0.23	0.05	0.75 (0.67–1.23)	0.16		
	Mussel	0.51	0.65	1.69 (0.67–13.23)	2.18		
Zinc	Squid	11.32	1.60	37.70 (24.90–52.73)	5.33	Male: 10,100 Female: 8200	40
	Oyster	202.60	88.41	675.33 (252.25–1688.13)	294.70		
	Mussel	24.96	20.04	83.17 (30.27–322.76)	67.62		

ND = no data; SE = standard error; RDA = recommended dietary allowance; UL = tolerable upper intake level.

## Data Availability

The data used to support the findings of this study can be made available by the corresponding author upon request.

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
