# Peer review of "To Eat or Not to Eat?—Food Safety Aspects of Essential Metals in Seafood"

_foods, 2023, doi:10.3390/foods12224082_

Round 1

Reviewer 1 Report

Comments and Suggestions for Authors

The manuscript is dedicated to the studies of metal contents in sea products (squid, oyster, mussel) of different origin purchased at the market. It is of practical value.

There are a few questions to be resolved:

1) What is the point in giving the average values for metal accumulation in different species of mussels (4 species) and oysters (2 species)?  It would have been more useful to provide the data per species.

2) Table 2: Please specify what is marked by asterisks?

In fact, as in the text of the article metal accumulation in squids, oysters, mussels is compared all the time, it would have been more appropriate to provide the lettering (abc) along with the values in the Table based on post-hoc statistical analysis to show the significant differences between the squids, oysters, mussels. Please provide.

3) Table 3: Please check the calculations. E.g., if Co content in mussel is 0.16 , then its average uptake can`t be 0.05. Also, if Cr content in squid is 0.06, its average uptake can`t be 0.98. And so on.

Comments on the Quality of English Language

Minor editing of English language is required.

Author Response

Responses to Reviewer 1

Remark 1: Does the introduction provide sufficient background and include all relevant references? Can be improved.

Response 1: This part has been revised and completed based on the proposal. Please see the revised text.

Remark 2: Are all the cited references relevant to the research? YES

Response 2: No response.

Remark 3: Is the research design appropriate? Can be improved.

Response 3: We investigated 3 groups of molluscs (gastropods, cephalopods) including 1 group of different type of shellfish/cramps/mussels (with symmetric shells), 1 group of oysters with asymmetric shells, and 1 group of cephalopods (squids). The samples were taken on a fishery market, the number of the type of molluscs was different based on the supply of the market.

The measured concentrations of the different investigated metals were analysed statistically among the 3 groups. Based on the detected amount of the investigated metals, the metal burden was calculated in all cases using an average consumption and an average consumer body weight, and then these calculated data were compared with the daily necessity and tolerable level of the metals.

To our opinion this research design is accepted, adequate and usual in food safety and/or food hygiene study to investigate the protection of consumer safety.

Remark 4: Are the methods adequately described? Can be improved.

Response 4: The Method section contains the steps of preparation of the samples including the details of equipment (tool, machine etc.) and the chemicals and standards which are used during the chemical analysis. Furthermore, the analytical method is precisely described including the measurement parameters. The validation process is also included in this section including the investigated parameters and their evaluation.

To our opinion the methods are adequately described. The description of the statistical analysis was modified a bit. Please see the revised text.

Remark 5: Are the results clearly presented? Can be improved.

Response 5: The results have been re-calculated based on the proposal. Please see the revised text.

Remark 6: Are the conclusions supported by the results? Can be improved.

Response 6: A separate Conclusion chapter is not necessary based on the „Instructions for Authors” of this journal. The conclusions are incorporated in the Discussion chapter at the different metals.

Remark 7: Comments and Suggestions for Authors

The manuscript is dedicated to the studies of metal contents in sea products (squid, oyster, mussel) of different origin purchased at the market. It is of practical value.

Response 7: Thanks for your opinion.

Remark 8: What is the point in giving the average values for metal accumulation in different species of mussels (4 species) and oysters (2 species)?  It would have been more useful to provide the data per species.

Response 8: We evaluated and compared the results of shellfish species and the oyster species. To our opinion this comparison is adequate since the shellfish and oysters are classified into the same kingdom, phylum, and class, however, the orders are different, thus there may difference between them e.g. in nutrition, physiology etc. This kind of grouping and evaluation is well-founded, which is generally clearly shown by the difference in the data between the two groups (shellfish and oysters); additional grouping suggested by the reviewer is probably not necessary.

Remark 9: Table 2: Please specify what is marked by asterisks?

In fact, as in the text of the article metal accumulation in squids, oysters, mussels is compared all the time, it would have been more appropriate to provide the lettering (abc) along with the values in the Table based on post-hoc statistical analysis to show the significant differences between the squids, oysters, mussels. Please provide.

Response 9: The Table 2 has been modified and completed with “compact letter display” (abc) based on the proposal. Please see the revised text/table.

Remark 10: Table 3: Please check the calculations. E.g., if Co content in mussel is 0.16 , then its average uptake can`t be 0.05. Also, if Cr content in squid is 0.06, its average uptake can`t be 0.98. And so on.

Response 10: The data have been re-calculated based on the proposal, and the results have been modified in the Table 3 and the text. Please see the revised text.

Remark 11: Comments on the Quality of English Language: Minor editing of English language is required.

Response 11: The manuscript has been revised by native English-speaking person. Please see the revised manuscript.

Reviewer 2 Report

Comments and Suggestions for Authors

I would like to thank, for the opportunity to review the text of the manuscript titled: To eat or not to eat? - Food safety aspects of essential metals in seafood. This is a research paper in which the authors decided to analyze the content of selected metals in selected seafood products. The results of the quantitative analyses were converted to the amount in a meal to verify whether seafood is a good/safe source of selected metals.

The work is interesting and the research done was well designed. Unfortunately, despite the very interesting results, the quality of the manuscript is poor. Two chapters need complete improvement: introduction and discussion. One chapter is missing: conclusions.

Specific comments:

title - please, instead of a marketing slogan, indicate a precise, scientific title of the work that makes the form of the study clear (the current title does not show whether it is a research or review paper).

Language: please revise the following throughout the manuscript:

- please do not use personal pronouns and the personal form of speech

- please verify the correctness of the notation of enzyme names, English and Latin names of the described species

- use of upper or lowercase letters when indicating the names of chemical elements

Abstract: please explain the abbreviation used in the abstract

Introduction:

lines: 36-37: delete. The aim of the study is presented after the theoretical introduction and not before.

Please carefully review the current scientific literature and describe which elements are currently indicated as microelements and which are not. Quoting publications from the 1970s and 1980s in this context is outdated and unacceptable. It is also a source of error due to the omission of the results of a number of studies that negate earlier observations. Please remove the sources: 27, 50, 51 and 52 as outdated works. Please remove the descriptions created on their basis and prepare new ones based on sources no older than 5-10 years.

When writing about dietary deficiencies, please emphasize the results in humans. If there are none, please clearly indicate which descriptions apply to the animal model. Writing that magnesium deficiency leads in first step to cranial malformations is surprising. Please point out the most common deficiency symptoms and possibly follow up with descriptions of the impact of profound deficiency in various research models.

Lines 132-135: delete. The inhalation route is by no means a dietary or supplementation route of magnesium administration. No such descriptions have been indicated for other elements, and it is completely unnecessary in this paper.

The sentence in line 146 has an unclear message.

Discussion:

at the moment, the chapter is merely a repetition of the results. Please prepare a correct discussion of the results obtained based on numerous studies on similar issues (preferably based on sources no older than 10 years).

No chapter: conclusions.

First of all, please be careful in using the terms: microelement, essential element. Please base the proper use of nomenclature for selected metals on current scientific literature.

Author Response

Responses to Reviewer 2

Quality of English Language

(x) I am not qualified to assess the quality of English in this paper

Comments and Suggestions for Authors

I would like to thank, for the opportunity to review the text of the manuscript titled: To eat or not to eat? - Food safety aspects of essential metals in seafood. This is a research paper in which the authors decided to analyze the content of selected metals in selected seafood products. The results of the quantitative analyses were converted to the amount in a meal to verify whether seafood is a good/safe source of selected metals.

Response: Thanks for your opinion.

General remark: The work is interesting and the research done was well designed. Unfortunately, despite the very interesting results, the quality of the manuscript is poor. Two chapters need complete improvement: introduction and discussion. One chapter is missing: conclusions.

Response: The Introduction and Discussion section have been revised and completed based on the proposal. However, the Conclusion section is not mandatory according to the „Instructions for Authors” of this journal.

Remark 1: Does the introduction provide sufficient background and include all relevant references? Must be improved.

Response 1: This part has been revised and completed based on the proposal. Please see the revised text.

Remark 2: Are all the cited references relevant to the research? YES

Response 2: No remark thus No response.

Remark 3: Is the research design appropriate? Can be improved.

Response 3: We investigated 3 groups of molluscs (gastropods, cephalopods) including 1 group of different type of shellfish/cramps/mussels (with symmetric shells), 1 group of oysters with asymmetric shells, and 1 group of cephalopods (squids). The samples were taken on a fishery market, the number of the type of molluscs was different based on the supply of the market.

The measured concentrations of the different investigated metals were analysed statistically among the 3 groups. Based on the detected amount of the investigated metals, the metal burden was calculated in all cases using an average consumption and an average consumer body weight, and then these calculated data were compared with the daily necessity and tolerable level of the metals.

To our opinion this research design is accepted, adequate and usual in food safety and/or food hygiene study to investigate the protection of consumer safety.

Remark 4: Are the methods adequately described? YES

Response 4: No remark thus No response.

Remark 5: Are the results clearly presented? Can be improved.

Response 5: The results have been re-calculated based on the proposal. Please see the revised text.

Remark 6: Are the conclusions supported by the results? Can be improved.

Response 6: A separate Conclusion chapter is not necessary based on the „Instructions for Authors” of this journal. The conclusions are incorporated in the Discussion chapter at the different metals.

Remark 7: title - please, instead of a marketing slogan, indicate a precise, scientific title of the work that makes the form of the study clear (the current title does not show whether it is a research or review paper).

Response 7: Sorry, but the title is not a marketing slogan. It is very important from point of view of food hygiene and food safety that the foods can be safely consumed (can be eaten) by the consumers and they cannot be contaminated with chemicals over the regulated official acceptable/tolerable level.

Remark 8: Language: please revise the following throughout the manuscript:

- please do not use personal pronouns and the personal form of speech

- please verify the correctness of the notation of enzyme names, English and Latin names of the described species

- use of upper or lowercase letters when indicating the names of chemical elements

Response 8: The manuscript has been revised based on the proposal. Please see the revised text.

Remark 9: Abstract: please explain the abbreviation used in the abstract.

Response 9: The abbreviated word ICP-OES has been explained based on the proposal. Please see the revised text.

Remark 10: Introduction: lines: 36-37: delete. The aim of the study is presented after the theoretical introduction and not before.

Response 10: The sentence line 36-37 has been deleted based on the proposal. Please see the revised text.

Remark 11: Introduction: Please carefully review the current scientific literature and describe which elements are currently indicated as microelements and which are not. Quoting publications from the 1970s and 1980s in this context is outdated and unacceptable. It is also a source of error due to the omission of the results of a number of studies that negate earlier observations. Please remove the sources: 27, 50, 51 and 52 as outdated works. Please remove the descriptions created on their basis and prepare new ones based on sources no older than 5-10 years.

Response 11: This part has been revised and completed with newer scientific articles based on the proposal. The articles before 1990 have been deleted and newer ones have been incorporated in the text based on the proposal, and the Discussion section have been revised as necessary. Please see the revised text.

Remark 12: Introduction: When writing about dietary deficiencies, please emphasize the results in humans. If there are none, please clearly indicate which descriptions apply to the animal model.

Response 12: This part has been revised and completed based on the proposal. Please see the revised text.

Remark 13: Introduction: Writing that magnesium deficiency leads in first step to cranial malformations is surprising. Please point out the most common deficiency symptoms and possibly follow up with descriptions of the impact of profound deficiency in various research models.

Response 13: Sorry, but we write about the health problems with manganese (Mn) and not with magnesium (Mg). Based on the scientific literature the manganese (Mn) can induce skeletal abnormalities in animals.

However, description of properties of manganese has been revised and re-written. Please see the revised text.

Remark 14: Introduction: Lines 132-135: delete. The inhalation route is by no means a dietary or supplementation route of magnesium administration. No such descriptions have been indicated for other elements, and it is completely unnecessary in this paper.

Response 14: The sentences in line 132-135 have been deleted based on the proposal. Please see the revised text.

Remark 15: The sentence in line 146 has an unclear message.

Response 15: The mentioned sentence in line 146 has been re-written. Please see the revised text.

Remark 16: Discussion: at the moment, the chapter is merely a repetition of the results. Please prepare a correct discussion of the results obtained based on numerous studies on similar issues (preferably based on sources no older than 10 years).

Response 16: Yes, our results are shortly summarized that to compare them with the international literature. However, these sentences have been modified at each metal based on the reviewer’s opinion.

Now, 19 articles are used to evaluate our results with the scientific literature.

Sorry, but the older articles are also true and valid. The majority of these articles (16 articles) was published after 2002 (2002-2017), and only 3 articles were originated from 1996-1997.

However, this chapter/part has been completed with 6 newer articles to compare our results with the scientific literature.

Remark 17: No chapter: conclusions.

Response 17: Sorry, but the conclusion section is not obligatory based on the „Instructions for Authors” of this journal.

Remark 18: First of all, please be careful in using the terms: microelement, essential element. Please base the proper use of nomenclature for selected metals on current scientific literature.

Response 18: The Introduction section has been revised based on the proposal using the proper terms for metals.
